The non-HDL-C to APOB ratio as a predictor of inaccurate LDL-C measurement in patients with chronic intrahepatic cholestasis and jaundice: a retrospective study

Cheng Yongjiang
Ye Jingyan
Huang Junyuan
Wang Yang wangyang90611@gzucm.edu.cn
Department of Clinical Laboratory, The First Affiliated Hospital of Guangzhou University of Chinese Medicine , Canton, Guangdong Province , China
Verger Alexis
Electronic publication date: 2024 Oct 4
Publication date: 2024
Volume: 12
Electronic Location ID: e18224
Received 2024 May 7; Accepted 2024 Sep 12
Copyright: © 2024 Cheng et al.
Copyright year: 2024
Copyright holder: Cheng et al.
License: This is an open access article distributed under the terms of the Creative Commons Attribution License, which permits unrestricted use, distribution, reproduction and adaptation in any medium and for any purpose provided that it is properly attributed. For attribution, the original author(s), title, publication source (PeerJ) and either DOI or URL of the article must be cited.
License URL: https://creativecommons.org/licenses/by/4.0/

Keywords: Cholestatic, Jaundice, Low-density lipoprotein cholesterol, High-density lipoprotein cholesterol, Non-HDL-C

Funding: The authors received no funding for this work.

==============================
Background

Cholestasis is characterized by the accumulation of bile in the liver or biliary system due to obstruction or impaired flow, necessitating lipid profiling to assess lipid metabolism abnormalities. Intrahepatic cholestasis, being the most significant type of cholestasis, further complicates the assessment of lipid abnormalities. However, the accuracy of low-density lipoprotein cholesterol (LDL-C) measurement in intrahepatic cholestasis patients remains uncertain.

Objective

This study aimed to evaluate the consistency of the homogeneous assay and the Friedewald formula in detecting LDL-C levels and identify factors influencing LDL-C test results in intrahepatic patients with cholestasis.

Methods

Retrospective analysis of laboratory data was conducted on intrahepatic cholestatic patients. Correlations between LDL-C values obtained using the homogeneous method (LDL-C(D)) and the Friedewald formula (LDL-C(F)), as well as associations between high-density lipoprotein cholesterol (HDL-C) and apolipoprotein A1 (ApoA1), LDL-C(D) and LDL-C(F), and apolipoprotein B (ApoB), were analyzed. Logistic regression analyses were employed to identify diagnostic indicators for inaccurate LDL-C measurements in intrahepatic cholestatic patients.

Results

Compared to patients with intrahepatic cholestasis without jaundice, the correlation between LDL-C(F) and LDL-C(D) was weaker in those with jaundice. Additionally, HDL-C exhibited a strong correlation with ApoA1 in both jaundice and non-jaundice cholestasis cases. Elevated non-HDL-C to APOB ratio (NH-C/B Ratio) levels (>4.5) were identified as a reliable predictor of inaccurate LDL-C measurements in patients with chronic intrahepatic cholestasis accompanied by jaundice.

Conclusions

LDL-C measurement reliability is moderately weaker in patients with intrahepatic cholestasis accompanied by jaundice. Elevated levels of the NH-C/B ratio serve as a significant predictor of inaccurate LDL-C measurements in this chronic patient population, highlighting its clinical relevance for diagnostic assessments.

Introduction

Cholestasis, characterized by the accumulation of bile in the liver or biliary system due to obstruction or impaired bile flow, is a significant medical condition. It can result from various factors such as bile duct obstruction or disturbances in bile flow (Brevini, Tysoe & Sampaziotis, 2020). Its impact on lipid metabolism, particularly in causing hypercholesterolemia, is a critical aspect that warrants detailed exploration. The relationship between cholestasis and hypercholesterolemia is multifaceted, potentially involving disruptions in the normal transport and metabolism of lipids within the liver (Longo, Crosignani & Podda, 2001). This interplay is not only clinically significant but also biologically intriguing, as it may involve alterations in key regulatory pathways such as those controlled by nuclear receptors and enzymes involved in bile acid synthesis and lipid homeostasis (Pawlak, Lefebvre & Staels, 2015).

Cholestasis syndromes can be classified based on their etiology into intrahepatic and extrahepatic cholestasis. Intrahepatic cholestasis represents the most prevalent form. When cholestasis endures beyond 6 months, it is designated as chronic cholestasis (European Association for the Study of the Liver, 2009). Lipid profiling plays a pivotal role in the assessment and management of patients with cholestasis, providing crucial information about serum lipid metabolism abnormalities (Nemes et al., 2016). Among the various lipid markers, low-density lipoprotein cholesterol (LDL-C) is of particular interest due to its association with cardiovascular risk and its importance in clinical decision-making processes (Drexel et al., 2021). The accurate measurement of LDL-C is paramount, especially in the context of cholestasis, where traditional methods may be compromised.

The current clinical methods for LDL-C measurement include the homogeneous assay and the Friedewald formula (Celik et al., 2010). The homogeneous assay encompasses various techniques such as the soluble method (Sol), surfactant method (SUR), protective method (PRO), catalase method (CAT), and ultraviolet method (CAL) (Momiyama et al., 2012). The Friedewald formula, developed in 1972 by Friedewald, Levy & Fredrickson (1972) is a simple formula that requires certain conditions. In a large community-dwelling population, it has been demonstrated that the homogeneous assay exhibits a strong correlation with the Friedewald formula in both fasting and nonfasting samples when triglyceride (TG) levels are below 4.52 mmol/L (Tanno et al., 2010).

Previous studies have reported that cholestasis can lead to secondary hypercholesterolemia (Longo, Crosignani & Podda, 2001). Accurately measuring LDL-C in patients with cholestasis poses unique challenges. Some studies have indicated that the presence of lipid abnormalities, including lipoprotein X(LP-X), can lead to inaccurate LDL-C test results (Fei et al., 2000; Heimerl et al., 2016; Kattah et al., 2019). This phenomenon underscores the need for a reevaluation of current diagnostic practices and the exploration of alternative methodologies that can provide more reliable results in this patient population.

In this study, we aimed to evaluate the consistency of the homogeneous assay and the Friedewald formula in detecting LDL-C levels in patients with intrahepatic cholestasis. Additionally, we assessed the factors influencing LDL-C test results by retrospectively collecting clinical data and laboratory parameters of the patients.

Materials and Methods

Patients

We collected data from the First Affiliated Hospital of Guangzhou University of Chinese Medicine, including patients admitted between January 2020 and January 2023, who were diagnosed with intrahepatic cholestasis. The patients were grouped according to specific inclusion criteria. The inclusion criteria were based on the 2009 guidelines for the diagnosis and management of cholestatic liver diseases published by European Association for the Study of the Liver (2009). These criteria included alkaline phosphatase (ALP) levels higher than 1.5 times the upper limit of normal (ULN) and gamma-glutamyl transferase (GGT) levels higher than three times the ULN. Additionally, cholestatic liver disease patients were further categorized into cholestatic jaundice group (total bilirubin, TB, level ≥ 51.3 µmol/L) and cholestatic non-jaundice group (TB level < 51.3 µmol/L). Our study excluded patients with renal insufficiency, defined as the presence of either acute kidney injury (AKI) or chronic kidney disease (CKD) as determined by clinical diagnosis and renal function tests. Additionally, we excluded pregnant or lactating women, individuals under 18 years of age, patients with incomplete data on prior biliary surgery, and those with triglyceride levels above 4.52 mmol/L or LDL-C levels below 1.81 mmol/L. To ensure our findings were not influenced by lipid-lowering treatments, we also excluded patients who had received such medications within the 6 months prior to the study.

A total of 652 intrahepatic cholestasis patients were included in the study, of which 118 were in the cholestatic jaundice group (63 males and 55 females) and 534 were in the cholestatic non-jaundice group (275 males and 259 females). The study was approved by the Ethics Committee of The First Affiliated Hospital of Guangzhou University of Chinese Medicine (K-2023-135) and was conducted in accordance with the Declaration of Helsinki. Depending on the nature of the research, the Ethics Committee of the First Affiliated Hospital of Guangzhou University of Chinese Medicine has waived the need for consent.

Data collection

Clinical data of hospitalized patients with cholestasis presenting for the first time were collected, including gender and age. Additionally, fasting venous blood samples of 3 mL were collected using Polyethylene Terephthalate vacuum procoagulant tubes. The collected blood samples were centrifuged at approximately 4,100 × g for 5 min using a medical centrifuge, and the resulting serum was used for laboratory testing of various parameters. All laboratory tests were performed using the ROCHE Cobas 8000 fully automated biochemical analyzer. The reagents used included the following kits provided by Roche Diagnostics (Shanghai, China) Co., Ltd.: Triglyceride Test Kit (GPO-PAP enzymatic colorimetric method), Cholesterol Test Kit (CHOD-PAP enzymatic colorimetric method), High-Density Lipoprotein Cholesterol Test Kit (selective inhibition method, PPD method), Low High-Density Lipoprotein Cholesterol Test Kit (solubilization method, SOL method), Gamma-Glutamyl Transferase Test Kit (γ-glutamyl-3-carboxy-4-nitroaniline rate method), and Alkaline Phosphatase Test Kit (AMP buffer rate method). The ApoA1/ApoB Test Kit (immunoturbidimetric method) was provided by North Control Bio-Tech Co., Ltd, (Sichuan, China). The Total Bilirubin Test Kit (vanadate method) and Direct Bilirubin Test Kit (vanadate method) were provided by Maccura Biological Technology (Sichuan, China) Co., Ltd.

Calculation of LDL-C(F)

The LDL-C measurement obtained through instrument analysis is referred to as LDL-C (D). LDL-C(F) is calculated using the Friedewald equation based on the instrument-measured values of TC, TG, and HDL-C. The Friedewald equation is as follows: LDL-C(F) = TC–HDL-C–TG/2.2. If the percentage difference ((LDL-C(D)–LDL-C(F))/LDL-C(D) is equal to or greater than 30%, we consider the results of LDL-C(D) and LDL-C(F) to be inaccurate. Conversely, if the percentage difference is less than 30%, we consider the results to be accurate.

Calculation of NH-C/B ratio

The non-HDL-C to APOB ratio (NH-C/B ratio) is calculated using the formula TC−HDL−CAPOB, where TC denotes total cholesterol, HDL-C denotes high-density lipoprotein cholesterol, and APOB represents apolipoprotein B.

Statistical analysis

Statistical analysis was performed using SPSS 25.0 software (SPSS, Armonk, NY, USA). For continuous variables, data were expressed as mean ± standard deviation if they followed a normal distribution, and as median (interquartile range) (M(Q1-Q3)) if they did not. When comparing two groups, if the data followed a normal distribution, the unpaired t-test was used for statistical analysis. If the data did not follow a normal distribution, a nonparametric method was employed. A two-tailed P-value of less than 0.05 was considered statistically significant. Linear correlation analyses were conducted between LDL-C(D) and LDL-C(F), HDL-C and ApoA1, LDL-C(D) and ApoB, and LDL-C(F) and ApoB in both the cholestatic jaundice and cholestatic non-jaundice groups. If the data followed a normal distribution, Pearson correlation analysis was used. If the data did not follow a normal distribution, Spearman correlation analysis was utilized.

Results

Intrahepatic cholestasis jaundice impacts LDL-C discrepancy

As shown in Fig. 1, a total of 652 cases of intrahepatic cholestasis were included in our study, with 118 cases in the cholestatic jaundice group and the remaining 534 cases in the cholestatic non-jaundice group. Furthermore, patients were categorized into acute and chronic groups based on whether the duration of cholestasis exceeded 6 months. Among the 118 cases with jaundice, there were 61 chronic and 57 acute cases; among the 534 cases without jaundice, there were 265 chronic and 269 acute cases. Laboratory test results for liver function and blood lipid parameters were collected for both groups. As shown in Table 1, there were significant differences in the laboratory test results of liver function and blood lipids between the two groups. Compared to the cholestatic non-jaundice group, the cholestatic jaundice group had higher levels of TC and TG, and lower levels of HDL-C. Interestingly, the proportion of patients in the cholestatic jaundice group with a discrepancy between LDL-C(F) and LDL-C(D) reached 36.4%, which was significantly higher than that in the cholestatic non-jaundice group.

Figure 1 Flow diagram of study population.

Based on the inclusion criteria, a total of 851 cases were enrolled. Of these, 199 cases of extrahepatic cholestasis were excluded, resulting in 652 cases of intrahepatic cholestasis. Among these 652 patients, 118 cases presented with jaundice based on total bilirubin (TB) levels, while 534 did not. Finally, patients were categorized into acute and chronic groups based on whether the duration of cholestasis exceeded 6 months. Among the 118 cases with jaundice, there were 61 chronic and 57 acute cases; among the 534 cases without jaundice, there were 265 chronic and 269 acute cases.

Table 1 Baseline characteristics of patients with intrahepatic cholestasis: comparative analysis with and without jaundice.

Characteristics	Cholestatic jaundice	Cholestatic non-jaundice	P value	
n	118	534		
Gender, n (%)			0.710a	
Male	63 (53.4%)	275 (51.5%)		
Female	55 (46.6%)	259 (48.5%)		
Age, median (IQR)	58 (51, 67)	59 (54, 67)	0.272b	
Type of cholestasis, n (%)			0.684a	
Chronic	61 (51.7%)	265 (49.6%)		
Acute	57 (48.3%)	269 (50.4%)		
apoA1, median (IQR)	0.71 (0.54, 0.87)	1.07 (0.84, 1.37)	<0.001b	
apoB, median (IQR)	1.10 (0.90, 1.43)	0.95 (0.81, 1.16)	<0.001b	
TC, median (IQR)	5.47 (4.26, 7.44)	4.66 (3.99, 5.61)	<0.001b	
TG, median (IQR)	1.50 (1.16, 2.02)	1.16 (0.86, 1.60)	<0.001b	
HDL-C, median (IQR)	0.59 (0.45, 0.86)	1.07 (0.79, 1.41)	<0.001b	
GGT, median (IQR)	501.0 (332.8, 818.5)	310.5 (226.3, 444.8)	<0.001b	
ALP, median (IQR)	400.0 (285.3, 619.3)	251.0 (200.0, 339.0)	<0.001b	
TB, median (IQR)	104.3 (73.4, 151.2)	10.9 (6.9, 18.8)	<0.001b	
DB, median (IQR)	70.9 (49.1, 108.8)	5.4 (3.0, 10.8)	<0.001b	
TBA, median (IQR)	75.4 (36.1, 145.1)	7.9 (4.0, 18.3)	<0.001b	
AST, median (IQR)	149.0 (94.5, 268.0)	52.0 (32.3, 83.8)	<0.001b	
ALT, median (IQR)	82.5 (49.0, 157.8)	36.0 (22.0, 59.8)	<0.001b	
LDL-C(D), median (IQR)	2.99 (2.41, 3.49)	3.00 (2.44, 3.72)	0.603b	
LDL-C(F), median (IQR)	3.92 (2.82, 5.77)	2.86 (2.34, 3.67)	<0.001b	
NH-C/B ratio, median (IQR)	3.99 (3.47, 4.61)	3.63 (3.38, 3.88)	<0.001b	
Lipid testing, n (%)			<0.001a	
Accurate	75 (63.6%)	482 (90.3%)		
Inaccurate	43 (36.4%)	52 (9.7%)		
Notes:

a Denotes the use of the chi-square test (Chisq test).

b Indicates the application of the Wilcoxon test.

Abbreviations: apoA1, Apolipoprotein A1; apoB, Apolipoprotein B; TC, Total cholesterol; TG, Triglycerides; HDL-C, High-density lipoprotein cholesterol; GGT, Gamma-glutamyl transferase; ALP, Alkaline phosphatase; TB, Total bilirubin; DB, Direct bilirubin; IB, Indirect bilirubin; TBA, Total bile acids; AST, Aspartate aminotransferase; ALT, Alanine aminotransferase; LDL-C(D), Low-density lipoprotein cholesterol (the homogeneous method); LDL-C(F), Low-density lipoprotein cholesterol (calculated using the Friedewald formula); NH-C/B ratio, the ratio of non-HDL-C and APOB ((TC-HDL-C)/APOB).

Weak LDL-C correlations in intrahepatic cholestasis jaundice

Furthermore, we conducted a correlation analysis on certain lipid profile parameters. As shown in Fig. 2A, we found a strong correlation between HDL-C and ApoA1 in both the cholestatic non-jaundice and cholestatic jaundice groups. However, as depicted in Fig. 2B, the correlation coefficient between LDL-C(F) and LDL-C(D) in the cholestatic jaundice group was 0.633, indicating a relatively weak correlation, whereas in the cholestatic non-jaundice group, these two variables exhibited a strong correlation. Further analysis revealed, as shown in Figs. 2C and 2D, a relatively weak correlation between LDL-C(F) and ApoB, as well as between LDL-C(D) and ApoB in the cholestatic jaundice group, with correlation coefficients significantly lower than those observed in the cholestatic non-jaundice group.

Figure 2 Correlation analysis of blood lipid parameters between the intrahepatic cholestasis non-jaundice and intrahepatic cholestasis jaundice groups.

(A) HDL-C and ApoA1 showed a strong positive correlation in both groups. (B)–(D) reveal relatively weak correlations between LDL-C(F) and LDL-C(D), LDL-C(F) and ApoB, as well as LDL-C(D) and ApoB in the jaundice group. In contrast, the non-jaundice group displayed strong correlations among these variables. Abbreviations: apoA1, Apolipoprotein A1; apoB, Apolipoprotein B; HDL-C, High-density lipoprotein cholesterol; LDL-C(D), Low-density lipoprotein cholesterol (the homogeneous method); LDL-C(F), Low-density lipoprotein cholesterol (calculated using the Friedewald formula).

NH-C/B ratio and TC as predictors for LDL-C accuracy

In order to identify additional predictors of inaccurate LDL-C measurements in patients with intrahepatic cholestatic jaundice, we conducted further analysis of laboratory data collected from 118 cases in the cholestatic jaundice group, as presented in Table 2. Univariate and multivariate logistic regression analyses revealed that TC, TG, ALP and NH-C/B ratio were independent markers for predicting inaccurate LDL-C measurements in patients with cholestatic jaundice.

Table 2 Predicting inaccuracies in LDL-C measurements among intrahepatic cholestatic patients with jaundice: a univariate and multivariate logistic regression analysis.

Characteristics	Total(N)	Univariate analysis		Multivariate analysis	
Odds ratio (95% CI)	P value	Odds ratio (95% CI)	P value	
Gender	118						
Male	63	Reference					
Female	55	1.151 [0.543–2.440]	0.714				
Age	118	1.019 [0.989–1.049]	0.224				
apoA1	118	0.093 [0.020–0.432]	0.002		0.065 [0.001–4.586]	0.208	
apoB	118	1.814 [0.762–4.318]	0.178				
TC	118	1.281 [1.093–1.503]	0.002		0.542 [0.332–0.885]	0.014	
TG	118	1.653 [1.058–2.581]	0.027		3.007 [1.014–8.913]	0.047	
HDL-C	118	0.051 [0.010–0.269]	<0.001		0.783 [0.010–58.998]	0.912	
GGT	118	1.001 [1.000–1.001]	0.137				
ALP	118	1.002 [1.001–1.004]	<0.001		1.002 [1.000–1.004]	0.036	
TB	118	1.015 [1.008–1.023]	<0.001		0.985 [0.946–1.026]	0.463	
DB	118	1.025 [1.014–1.036]	<0.001		1.037 [0.975–1.103]	0.250	
TBA	118	1.006 [1.002–1.011]	0.005		1.005 [0.998–1.011]	0.139	
AST	118	1.000 [0.997–1.003]	0.889				
ALT	118	1.000 [0.997–1.004]	0.863				
NH-C/B ratio	118	1.916 [1.291–2.843]	0.001		2.952 [1.359–6.409]	0.006	
Note:

Variables with a P-value less than 0.1 in univariate analysis were included in the multivariate logistic regression analysis.

Abbreviations: apoA1, Apolipoprotein A1; apoB, Apolipoprotein B; TC, Total cholesterol; TG, Triglycerides; HDL-C, High-density lipoprotein cholesterol; GGT, Gamma-glutamyl transferase; ALP, Alkaline phosphatase; TB, Total bilirubin; DB, Direct bilirubin; TBA, Total bile acids; AST, Aspartate aminotransferase; ALT, Alanine aminotransferase; NH-C/B Ratio, the ratio of non-HDL-C and APOB ((TC-HDL-C)/APOB).

Furthermore, we conducted ROC curve analysis to evaluate the diagnostic performance of TC, TG, ALP and NH-C/B ratio as predictors of inaccurate LDL-C measurements in patients with intrahepatic cholestatic jaundice. As shown in Fig. 3A, the results indicated that TC and NH-C/B ratio demonstrated relatively good diagnostic efficacy, with areas under the curve (AUCs) of 0.770. To further investigate the diagnostic performance of these two parameters in acute and chronic intrahepatic cholestasis, we conducted separate analyses. The results, as shown in Fig. 3B, revealed that in chronic intrahepatic cholestasis, the NH-C/B ratio showed relatively better diagnostic efficacy compared to TC, with an AUC of 0.770 (95% CI [0.641–0.900]). Utilizing a cut-off value of 4.50 for the NH-C/B ratio, the sensitivity was 62.5% and the specificity was 83.8%. On the other hand, as depicted in Fig. 3C, in acute intrahepatic cholestasis, TC exhibited relatively better diagnostic efficacy compared to the NH-C/B ratio, with an AUC of 0.818 (95% CI [0.709–0.927]). With a cut-off value of 5.03 mmol/L for TC, the sensitivity was 94.7% and the specificity was 68.4%.

Figure 3 ROC analysis of TC, TG, ALP, and NH-C/B ratio for predicting LDL-C measurement inaccuracies in intrahepatic cholestasis with jaundice.

(A) presents patients with Intrahepatic Cholestasis and jaundice, including both acute and chronic cases. (B) specifies to chronic Intrahepatic Cholestasis patients with jaundice. (C) specifies to acute Intrahepatic Cholestasis cases with jaundice. Abbreviations: TC, Total cholesterol; TG, Triglycerides; ALP, Alkaline phosphatase; NH-C/B ratio, the ratio of non-HDL-C and APOB ((TC-HDL-C)/APOB).

Discussion

Blood lipid testing is commonly performed in clinical practice for patients with cholestasis, and obtaining accurate measurement results is crucial for effective management of lipid abnormalities. Additionally, there are studies suggesting that lipid metabolism abnormalities contribute to the pathogenesis of cholestasis (Fu et al., 2019; Gao et al., 2022), further highlighting the significance of accurate blood lipid testing in these patients. In our study, we found that in patients with intrahepatic cholestasis accompanied by jaundice, the measurement results of LDL-C were inaccurate. Compared to patients without jaundice, the rate of discordance between the homogeneous assay (Sol method) and the Friedewald formula significantly increased, indicating that jaundice may be a key factor interfering with LDL-C measurement. Lp-X is an abnormal lipoprotein type most commonly seen in patients with liver dysfunction, specifically in those patients with intra- or extrahepatic cholestasis (Ashorobi & Liao, 2024; Heimerl et al., 2016). However, due to limitations in our study, we did not assess the presence of lipoprotein-X (LP-X), and further investigations are planned to delve deeper into this aspect. To further validate our findings, we conducted a study on the correlation between HDL-C, LDL-C, and their respective apolipoproteins in two groups of patients: those with cholestasis accompanied by jaundice and those without jaundice. The results demonstrated a strong correlation between HDL-C and ApoA1 in both groups, indicating the reliable measurement results of HDL-C using the homogeneous assay (PPD method) that are unaffected by cholestasis and jaundice. In the group without jaundice, LDL-C(D) exhibited a significant correlation with ApoB, as did LDL-C(F) with ApoB. However, in the intrahepatic cholestatic jaundice group, the correlations between LDL-C(D) and ApoB, as well as between LDL-C(F) and ApoB, were weaker. Considering that the detection of apolipoproteins involves immunoturbidimetry, which has high specificity, while the homogeneous assay for LDL-C is influenced by multiple factors (Iwasaki, Matsuyama & Nakashima, 2006), these findings further suggest the unreliability of LDL-C(D) and LDL-C(F) measurements in patients with cholestatic jaundice.

Previous research has predominantly concentrated on diagnostic and prognostic markers for cholestasis, GGT and ALP have been established as such markers (Chen et al., 2021; Lu, 2022). In our study, we identified TC, TG, ALP and NH-C/B ratio as independent indicators for predicting inaccurate LDL-C measurements in patients with intrahepatic cholestatic jaundice. Among these, TC and NH-C/B ratio demonstrated the highest diagnostic efficacy. Stratifying cholestasis is crucial for accurate lipid profiling, especially in chronic cases for cardiovascular risk management. In our evaluation of patients with intrahepatic cholestasis and jaundice, stratified by chronicity, the NH-C/B ratio was found to be more diagnostically efficacious than TC. This ratio shows promise as a clinical metric, with evidence implicating the non-HDL-C/apoB ratio as a prognostic biomarker for long-term mortality risks, independent of traditional risk factors (Zhang et al., 2024). Notably, an NH-C/B ratio threshold of 4.5 in chronic cases signifies the potential for inaccurate LDL-C measurements, underscoring the necessity for alternative diagnostic strategies. Our study’s findings highlight the imperative for refined LDL-C assessment techniques to enhance diagnostic accuracy in chronic intrahepatic cholestasis with jaundice.

However, it is important to acknowledge several limitations of our study. Firstly, the retrospective nature of the data collection may introduce inherent biases. Additionally, the relatively small sample size and the lack of longitudinal follow-up may restrict the generalizability of our findings. Future research should aim to validate these findings in larger, more diverse cohorts and explore the underlying mechanisms driving these alterations in lipid profiles.

Conclusions

In conclusion, this study highlights a diminished correlation between LDL-C measurements obtained through the homogeneous assay and the Friedewald formula in patients with intrahepatic cholestatic jaundice. These findings indicate that LDL-C measurements using both methods are unreliable in patients with intrahepatic cholestasis accompanied by jaundice. Furthermore, our study underscores the significance of the non-HDL-C to APOB ratio (NH-C/B ratio) as a predictor for inaccurate LDL-C measurements in patients with chronic intrahepatic cholestasis accompanied by jaundice. Elevated NH-C/B ratio levels (>4.5) serve as an indicator of unreliable LDL-C testing.

Supplemental Information

Supplemental Information 1 Raw data.

Supplemental Information 2 STROBE checklist.

Additional Information and Declarations

Competing Interests

Author Contributions

Human Ethics

Data Availability

The authors declare that they have no competing interests.

Yongjiang Cheng performed the experiments, analyzed the data, prepared figures and/or tables, authored or reviewed drafts of the article, and approved the final draft.

Jingyan Ye performed the experiments, analyzed the data, prepared figures and/or tables, and approved the final draft.

Junyuan Huang analyzed the data, authored or reviewed drafts of the article, and approved the final draft.

Yang Wang conceived and designed the experiments, analyzed the data, prepared figures and/or tables, authored or reviewed drafts of the article, and approved the final draft.

The following information was supplied relating to ethical approvals (i.e., approving body and any reference numbers):

The Ethics Committee of The First Affiliated Hospital of Guangzhou University of Chinese Medicine granted Ethical approval to carry out the study within its facilities (Ethical Application Ref: K-2023-135).

The following information was supplied regarding data availability:

The raw data is available in the Supplemental FIle.

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
