# Peer review of "The non-HDL-C to APOB ratio as a predictor of inaccurate LDL-C measurement in patients with chronic intrahepatic cholestasis and jaundice: a retrospective study"

_PeerJ, doi:10.7717/peerj.18224_

## Round 0.1 · original submission · Major Revisions

Your manuscript has now been seen by 3 referees. As you will see from their comments (attached below) they find this work of potential interest, but have raised substantial concerns, which in our view would need to be addressed with major revisions before we can consider publication.

I should stress that the referees’ concerns point to the absence of stratification of cholestasis type and duration within the cohort and the lack of exclusion criteria precision which would need to be addressed.

We would be happy to consider a revised manuscript that would satisfactorily address all the points raised by the referees.

We hope that you will find referees' comments and editorial guidance helpful.

Regards,

Alexis Verger

Reviewer 1 ·

Basic reporting

Minor comments:
Abstract:
1. Line 47:CHOL is an abbreviation for cholesterol,please change the abbreviation for total cholesterol to TC, and modify the entire text
Introduction:
1. line 68: “triglyceride” change to “triglyceride (TG)”.
Materials & Methods:
1. line 92 “triglyceride (TG)” change to “TG”
2. line 103: Please write the full English name of PET
Discussion:
1. Line 189-191: “Conversely, to the best of our knowledge, there is currently no literature available that documents the inaccurate measurement results of LDL-C in patients with cholestasis accompanied by jaundice.” This statement is not accurate. LPx may be present in patients with cholestasis. In fact, LDL-C calculated using the Friedewald formula is invalid in the presence of LpX. Moreover, LpX can interfere with direct LDL-C assays, the degree of interference being dependent on the method used【PMID:25286006】. In addition, there is no evidence to suggest that LpX can increase the risk of cardiovascular events, so for patients with cholestasis, direct testing of APOB is more valuable than testing of LDL-c.
Table:
1. Table 1 and table 2: please change “P value” to “P value”【Use an italic font】
2. Table 1: please unify the number of decimal places for each row

Experimental design

1. The reliability of the Friedewald formula calculation results is poor under extremely low LDL-C (<1.81mmol/L). In both group, there were some patients with hypobetalipoproteinemia due to chronic liver disease or familial hypobetalipoproteinemia. It is recommended to exclude these patients before further comparison as a supplement.
2.Table 1 & Table 2: It is recommended to incorporate the ratio of non-HDL-C and APOB [(TC-HDL-C)/APOB] into the analysis.

Validity of the findings

no comment

Reviewer 2 ·

Basic reporting

The english is clear and unambiguous, but should be improved.

Article structure and Reference is sufficient

Experimental design

Original primary research within Aims and Scope of the journal.

The submission clearly define the research question.

Methods described with sufficient detail..

Validity of the findings

All underlying data have been provided.

The data on which the conclusions should be improved.

Additional comments

None

·

Basic reporting

The article addresses an important yet underexplored issue, and the authors' purpose is clearly articulated from the first line. The writing is simple and understandable, enhancing the comprehension of the topic. I suggest the following modifications to improve the English and clarity of the text:

- Line 153: Change to "Furthermore, we conducted a correlation analysis on certain lipid profile parameters."

- Lines 156-157: The phrase "…correlation coefficient between LDL-C(F) and LDL-C(D) in the cholestatic jaundice group was 0.507, indicating a moderate and relatively weak correlation, whereas in the cholestatic" can be confusing due to the use of both "moderate" and "relatively weak." Consider rephrasing it with only "relatively weak"

- Figure 1: Modify the title to "Correlation analysis of blood lipid parameters between the cholestatic non-jaundice and cholestatic jaundice groups."

Introduction: Enhance the focus on the relationship between cholestasis and hypercholesterolemia. Briefly outline the hypothetical mechanisms that could be altered in this phenomenon.

Acronym Usage: Re-evaluate the use of acronyms to avoid repetition and enhance clarity:

- Line 72: Introduce "lipoprotein X" before using the acronym LP-X.
Methods Section: Avoid repeating acronyms when explaining techniques. For example, in lines 109-110, rephrase to "High-Density Lipoprotein Cholesterol (HDL-C) Test Kit (selective inhibition method, PPD method)."
- Line 120: Use a single acronym for total cholesterol. If you choose "CHOL" ensure consistent use throughout the text without using a second one like "TC"
Line 167: Explain the acronym TBA the first time it is used.
Line 187: Ensure "lipoprotein-X (LP-X)" is introduced with the full term before using the acronym, ideally in line 72.

References: Update old citations and replace those that are not directly related to the topic with more specific literature:

Line 70: Reference 8 by Klose focuses on familial hypercholesterolemia, which may not be directly relevant. Cite articles specifically related to cholestasis and hypercholesterolemia instead.
Lines 204-205: Reference 17 by Choi et al is about the pediatric population, which may not be the best fit. Find references that discuss diagnostic and prognostic markers for cholestasis in the general population.
Line 180: Reference 2 on the metabolomics of ICP may be too specific. Consider a more general reference.
The paper "Cholesterol metabolism in cholestatic liver disease and liver transplantation: From molecular mechanisms to clinical implications" by Nemes et al. (2016) could be a useful addition to expand the introduction on the role of cholestasis.

These suggestions should improve the clarity and focus of the article, making it more suitable for publication.

Experimental design

While the study design is generally well-structured, several methodological issues need to be addressed to enhance the robustness and clarity of the study. Here are my specific concerns and suggestions:

1. Stratification of Cholestatic Diseases:
- Intrahepatic vs. Extrahepatic Cholestasis: The absence of stratification between intrahepatic and extrahepatic cholestasis is a critical oversight. These two types of cholestasis have different pathophysiological mechanisms and management approaches, which can significantly impact lipid metabolism.
- Chronic vs. Temporary Cholestasis: Differentiate between patients with temporary (e.g., obstructive forms that can be treated) and chronic cholestasis (lasting at least six months). Including this information and conducting dedicated sub-analyses will strengthen your findings.
Definition of Jaundice:
- Presence of Jaundice: It is essential to clarify whether all cases of jaundice are liver-related or if there are instances of hematologic jaundice. Ensure that the study only includes patients with direct bilirubin elevation to maintain consistency. Moreover the definition of jaundice as a total bilirubin level higher than 34.2 μmol/L (lines 88-90 ) should be clearly cited. Refer to your 2021 Guidelines for the Management of Cholestatic Liver Diseases to provide a reliable source.

2. Treatment for hypercholesterolemia:
Clarify whether the patients included in the study were already receiving treatment for dyslipidemia. This information is crucial as lipid-lowering treatments can significantly affect lipid profile results. Include this detail in the analysis to account for its potential impact.

3. Threshold for Inaccuracy:
Lines 121-122: The use of a 30% discordance threshold for defining inaccuracy is interesting but needs justification. Provide references or literature that support the choice of this specific threshold. If it is an arbitrary decision, discuss the rationale behind it.

4.Exclusion Criteria - Renal Insufficiency:
Line 90: The exclusion criteria mention renal insufficiency but lack a clear definition. Specify whether this includes acute kidney injury (AKI), chronic kidney disease (CKD), or both. Providing precise criteria will ensure that the study's exclusion parameters are well understood and reproducible

Validity of the findings

The inaccuracy of LDL measurement in cholestatic patients has a significant impact on managing lipid profile changes. This issue is particularly critical because the inability to accurately determine cardiovascular risk based on LDL levels hinders the ability to propose the best treatment options for patients with a history of cardiac diseases or metabolic syndrome.

Furthermore, the necessity to stratify different forms of cholestasis is essential. Understanding the etiology and chronicity of cholestasis and jaundice is vital for accurate lipid profile assessment. Patients with chronic cholestasis are the ones who truly need a redefined strategy for lipid assessment. These patients require consistent and accurate monitoring to manage their cardiovascular risk effectively.

In contrast, patients with acute or subacute forms of cholestasis, which are generally reversible, do not require lipid profile assessment until the condition resolves. Conducting lipid profile assessments during these transient phases may lead to misleading results and unnecessary interventions.

Surely the number of patients you provide is enough to reach a good relevance for your finding. However, to improve the validity of them, your work needs to be specifically targeted (if already not done) towards patients with chronic cholestasis, underlining the different type of cholestasis and the impact of LDL inaccuracy on the specific categories. This focus will ensure that the lipid assessment strategies are relevant and beneficial to the patients who need them the most. Addressing the inaccuracies in LDL measurement in this cohort will significantly enhance the management of their cardiovascular health.

Moreover, it would be beneficial to expand the discussion on the possible impact of the results, suggesting new research on more reliable methods for LDL measurement. Additionally, you could provide suggestions on the management of patients for example with cholesterol levels above 4.835 mmol/L.

---

## Round 0.2 · accepted · Accept

Dear Dr Wang,

in the light of the modifications you have taken into account following the reviewers' reports, I am happy to accept your revised manuscript "The Non-HDL-C to APOB Ratio as a Predictor of Inaccurate LDL-C Measurement in Patients with Chronic Intrahepatic Cholestasis and Jaundice : A Retrospective Study".

Sincerely,

Dr Alexis Verger